# The Electrophoretic Deposition of Nanopowders Based on Yttrium Oxide for Bulk Ceramics Fabrication

Elena Kalinina [1,2,3] and Maxim Ivanov [1,4,*]

1. Laboratory of High Purity Optical Materials, G.G. Devyatykh Institute of Chemistry of High-Purity Substances, Russian Academy of Sciences, 49 Tropinin Str., Nizhny Novgorod 603034, Russia
2. Laboratory of Complex Electrophysic Investigations, Institute of Electrophysics, Ural Branch of the Russian Academy of Sciences, Yekaterinburg 620016, Russia
3. Department of Physical and Inorganic Chemistry, Institute of Natural Sciences and Mathematics, Ural Federal University, Yekaterinburg 620002, Russia
4. Nonlinear Optics Laboratory, Institute of Electrophysics, Ural Branch of the Russian Academy of Sciences, Yekaterinburg 620016, Russia
* Correspondence: max@iep.uran.ru

**Abstract:** In the present work, a study was carried out to investigate the key factors that determine the uniformity, mass, thickness, and density of compacts obtained from nanopowders of solid solutions of yttrium and lanthanum oxides (($La_xY_{1-x}$)$_2O_3$) with the help of the electrophoretic deposition (EPD). Nanopowders were obtained by laser ablation of a mixture of powders of yttrium oxide and lanthanum oxide in air. The implemented mechanisms of the EPD and factors of stability of alcohol suspensions are analyzed. It has been shown that acetylacetone with a concentration of 1 mg/m$^2$ can be used as a dispersant for stabilization of isopropanol suspensions of the nanoparticles during the EPD. It was shown that the maximum density of dry compacts with a thickness of 2.4 mm reaches 37% of theoretical when EPD is performed in vertical direction from a suspension of nanopowders with addition of acetylacetone.

**Keywords:** electrophoretic deposition (EPD); nanoparticles; suspension; yttrium oxide; zeta potential

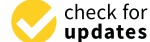



## 1. Introduction

One of the most important stages in the fabrication of optical ceramics is the compaction of nanoparticles to obtain bulk, mechanically unstressed compacts (green body) with a uniform density distribution [1]. It's well known that sintering activity of the nanoparticles increases with smaller size. However, surface energy of finer particles gives rise to stronger agglomeration, and homogeneity of the green body deteriorates with smaller particle sizes [2]. Upon dry pressing, the nanoparticles are shielded by the surrounding agglomerates and the applied pressure is not able to arrange the particles in optimum position within the three-dimensional array. This problem can be overcome with the help of colloidal methods, among which, first of all, slip casting was developed [3]. The use of colloidal methods for the formation of bulk compacts eliminates a number of limitations and disadvantages of pressing technology associated with defectiveness, porosity, internal stresses, and insufficient homogeneity of the obtained ceramic samples. The advantages of the slip casting include the possibility of obtaining compacts of various shapes and large geometric dimensions, its manufacturability and low cost. The disadvantages are the difficulty in selecting dispersants, and the contamination of ceramics with the remains of the dispersants that do not burn out during sintering, and with the material of the mold into which the casting is made. In fact, essential requirements of the slip casting—the lowers viscosity of the slurries combined with highest solid contents, make the technology widely used for submicron particles [3], but quite challenging in a case of the particles with a characteristic size of tens of nanometers [4].

One of the colloidal methods, where the requirements for the viscosity of suspensions and the concentration of the nanoparticles are not so strong, is electrophoretic deposition (EPD). Initially, EPD was developed for the formation of films [5], layers and coatings from nanoparticles [6]. This method makes it easy to control the thickness, morphology, and microstructure of coatings by changing the deposition time, applied voltage, and/or current [5,7–9], but to be successful the EPD requires the correct choice of a dispersion medium that solvates the particle surface well and ensures the aggregation and sedimentation stability of the suspension. Usually, alcohols and ketones, as well as their mixtures, are used as a dispersion medium [10–12]. A necessary condition for EPD is the appearance of an electric potential on the surface of nanoparticles dispersed in a liquid medium. The appearance of an electric charge is due, for example, to processes such as surface dissociation and adsorption of ions, which causes the formation of a double electric layer near the surface of nanoparticles and the appearance of mobility of nanoparticles under the action of an external electric field, which is characterized by the zeta potential value [13]. The greater the zeta potential, the greater the charge on the surface of the particle, which is a favorable factor in increasing the efficiency of the EPD. The choice of EPD modes and dispersants used to stabilize nanoparticles in suspension depends primarily on the dispersion medium.

Few examples of the application of the EPD for compacting bulk green body of ceramic materials, including the formation of optically transparent ceramics are presented in scientific papers. For instance, in [14] the EPD was carried out from aqueous dispersions of alumina, yttria, yttrium aluminum garnet and lutetium aluminum garnet with a characteristic size of the particles of about 100 nm. Films as well as cylindrical bodies were prepared by application of pulsed current, which was used to reduce the probability of formation of gas bubbles in the compact resulting from electrolysis of water at electric field up to 35 V/cm. Some works describe the preparation of ceramic compacts using non-aqueous suspensions [15,16] to avoid electrolysis, where the authors used organic dispersants and binders. During the sintering of the compacts, organic compounds burnt out with the release of gaseous products and the formation of pores in the obtained samples of sintered ceramics. Development of the EPD method to fabricate bulk green body compacts was demonstrated in [17], where it was proposed to obtain high-density optically transparent $Yb^{3+}$:$(La_xY_{1-x})_2O_3$ ceramics, which is one of promising laser materials [18–20], from non-aqueous suspensions (isopropyl alcohol) of weakly agglomerated nanoparticles with a characteristic size of 10–20 nm. It was shown that ceramics of high optical quality can be sintered from these fine nanoparticles compacted into a green body of relatively low density (less than 40%).

We believe that the EPD technology based on the use of non-aqueous dispersion media is quite promising, since it gives a chance to reduce gas evolution during the deposition process and thus reduce number of defects in ceramics. In this work, we investigated the EPD of green body from suspensions of $(La_xY_{1-x})_2O_3$ (YLO) nanoparticles with a characteristic size of 10–20 nm in isopropyl alcohol with a small amount of acetylacetone, which was used as a dispersant. The main objectives of the work were to determine the effect of the dispersant, suspension aging on the zeta potential and pH of the suspensions, deposition time and rate, the magnitude of the current, the mass of precipitates, the thickness and density of dry green body.

## 2. Results

Specific surface of YLO nanopowder measured by BET method was $S_{BET} = 76 \text{ m}^2/\text{g}$. Average size of nanoparticles calculated from $S_{BET}$ was 16 nm. The XRD analyses revealed the YLO nanoparticles to be single-phase solid solutions with monoclinic structure ($C2/m$), which is common for yttria nanoparticles produced by the LA [21,22]. In accordance with TEM (Figure 1a) the YLO nanopowders consist of weakly agglomerated spherical nanoparticles with average size of 15 nm that is typical for nanopowders made by LA [23]. Size distribution by number of the nanoparticles is shown in Figure 1b.

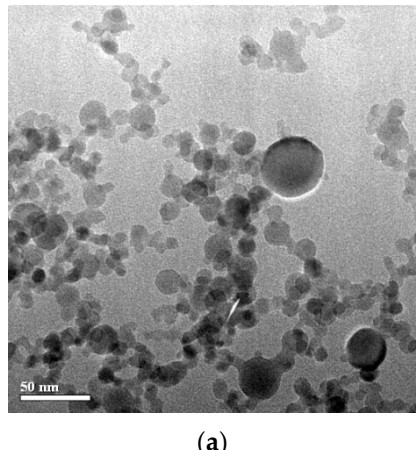 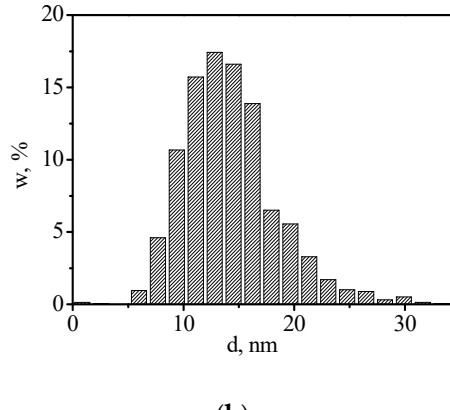

(**a**)           (**b**)

**Figure 1.** (**a**) TEM image of the $(La_xY_{1-x})_2O_3$ nanoparticles and (**b**) the particles size distribution by number.

*Methodology and Sequence of the Experiment to Obtain YLO Compacts.*

(Characteristics of suspensions and settling parameters are given in Table 1):

1.  Initial suspensions of YLO powder were prepared, they were ultrasonicated and deaggregated, YLO suspension in isopropyl alcohol is named YLO_i_1, YLO suspension in isopropyl alcohol with the addition of acetylacetone is named YLO_iA_1.
2.  Right after the deaggregation, EPD was carried out from YLO_i_1 and YLO_iA_1 suspensions in two variants of the deposition cell—with either vertical or horizontal EPD.
3.  Aging of the original suspension YLO_iA_1 for 2 months was made. The suspension was named YLO_iA_2. A vertical EPD was carried out from the YLO_iA_2 suspension and the remained suspension was named YLO_iA_3.
4.  The EPD was carried out from YLO_iA_3 suspension.

**Table 1.** Characteristics of the YLO suspensions and EPD conditions.

| No. | Suspension | Medium | pH | | $\zeta$-Potential, mV | | EPD Direction | Average Current, mA | Weight Energy Consumption |
| --- | --- | --- | --- | --- | --- | --- | --- | --- | --- |
| | | | Before EPD | After EPD | Before EPD | After EPD | | | |
| 1 | YLO_i_1 | iPrOH | 7.8 | 6.9 | +24 | +28 | Vertical | 0.005 | 91.4 mg 2031 mg/C |
| 2 | YLO_i_1 | iPrOH | 7.8 | 6.7 | +24 | +27 | Horizontal | 0.008 | 188.6 mg 2619 mg/C |
| 3 | YLO_iA_1 | iPrOH_Hacac | 7.9 | 6.7 | +83 | +104 | Vertical | 0.08 | 395 mg 549 mg/C |
| 4 | YLO_iA_2 | iPrOH_Hacac | 6.0 | 5.9 | +50 | +49 | Vertical | 0.136 | 402.3 mg 326 mg/C |
| 5 | YLO_iA_3 | iPrOH_Hacac | 5.9 | 5.9 | +39 | +39 | Vertical | 0.119 | 329.8 mg 307 mg/C |

It can be seen from Table 1 that, prior to EPD, the initial suspensions of different compositions of the dispersion medium are characterized by a high pH value and a rather high initial value of the $\zeta$ potential, especially for the suspension with the addition of acetylacetone, where the zeta potential is 3.5 times higher (+83 mV) compared to suspension in pure alcohol. When acetylacetone is added to a suspension of nanoparticles, the acetylacetone molecule interacts with the surface of the nanoparticles, which is accompanied by complexation with the release of free H+ protons, which, in turn, can protonate the surface of the nanoparticles, increasing the zeta potential of the nanoparticles. The

complex formation and generation of free protons can be represented by the following reaction: $M^{x+} + x\,Hacac \rightleftharpoons M(acac)_x + x\,H^+$ [24].

According to theoretical concepts and numerous experimental data, the suspension to be stable have to have the absolute value of the zeta potential that exceeds 20 mV either positive or negative [25]. This criterion is fulfilled in all the suspensions we studied (Table 1).

It turned out that the EPD process significantly affects the aggregative stability of the suspensions. In the works [11,26,27] it was shown that suspensions of oxide nanoparticles obtained by laser synthesis are highly stable in isopropyl alcohol as a result of so called "self-stabilization" process. In the initial YLO_i_1 suspension, the size distribution of particles (aggregates of nanoparticles) is shown in Figure 2a. It should be noted that the particle size ($d_{DLS}$) in suspension, measured using the dynamic light scattering (DLS) method, does not match the particle size determined using TEM. It was found that from the very beginning of the EPD process, intense flocculation of nanoparticles occurs, leading to a significant sedimentation of the suspension. Along with the initial aggregates with a size of hundreds of nanometers, aggregates with a size of a few micrometers appear (Figure 2a) and these large aggregates quickly sediment into the lower part of the cuvette. This flocculation isn't reversable and stopping the EPD process does not lead to deagglomeration of the nanoparticles.

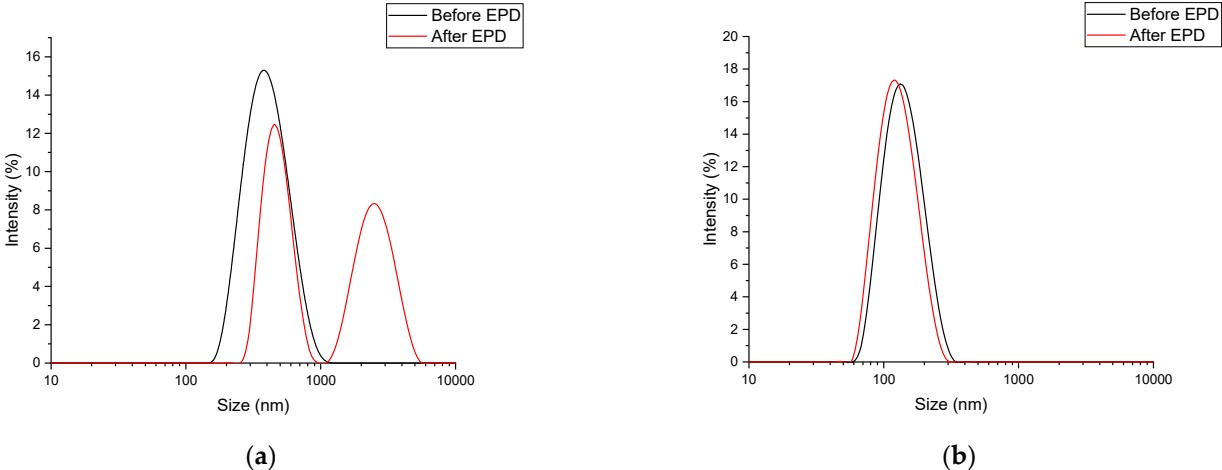

**Figure 2.** Size distribution by volume in (**a**) YLO_i_1 and (**b**) YLO_iA_1 suspensions before and after EPD.

The addition of acetylacetone significantly changes behaviour of the nanoparticles in the suspension. It not only reduces the size of the aggregates in the original suspension, but also prevents flocculation during the EPD process. Suspensions with the acetylacetone remained stable during the entire EPD process (Figure 2b).

In most of the experiments carried out, the pH value decreases during EPD. Table 1 shows that the change in pH during EPD from freshly prepared suspensions corresponds to an increase in the concentration of protons in the suspension by approximately 16 times (pH = 7.9 corresponds to a proton concentration of $1.258 \times 10^{-8}$ mol/L and pH = 6.7 corresponds to $1.995 \times 10^{-7}$ mol/L). One can estimate the corresponding value of the Debye length $\lambda_D$ in both cases by the equation [28]:

$$K = \frac{1}{\lambda_D} = \left( \frac{e^2 \sum_i n_i z_i^2}{\varepsilon \varepsilon_0 k_B T} \right)^{1/2} \qquad (1)$$

where $K = \frac{1}{\lambda_D}$—reciprocal of Debye length, 1/m; $e_0$—electron charge, C; $n_i$—concentration of ions with valency $z_i$, 1/m$^3$; $\varepsilon$—permittivity of the suspension; $\varepsilon_0$—dielectric constant; $k_B$—Boltzmann constant, J/K; $T$—absolute temperature, K.

Assuming the liquid medium is a 1-1 charge (symmetrical) electrolyte, for pH = 7.9, we obtain the Debye length value of 1.54 μm, and for pH = 6.7–0.38 μm. It can be seen that after the EPD process, a significant decrease in the Debye length occurs due to an increase in the concentration of protons in the suspension. At the same time, the Debye length significantly exceeds the average particle size of dry powder (d$_{TEM}$) but becomes comparable with the sizes of aggregates in suspension (d$_{DLS}$).

An increase in the concentration of protons in a liquid medium during EPD can be explained by the next processes:

1. In [29,30], a mechanism of electrochemical reactions in a suspension containing traces of water is described. It should be noted that in our case, according to TG-DSC data, the content of water adsorbed in the YLO nanopowders was about 8 wt. %. Taking into account the concentration of the nanopowder in the suspension, the water content added to isopropanol together with the powder could reach 5 g/L. In this medium, under the action of an electric field, not only positively charged particles move towards the cathode but the protons also, where they are reduced and molecular hydrogen is released. At the cathode, the local concentration of the protons decreases and pH increases. On the other hand, an electrochemical reaction occurs at the anode with the participation of water, as a result of which the protons and molecular oxygen are released into the solution that increases the concentration of protons and lowers the pH. Changes in the pH during the EPD observed in our experiments can be explained by a similar mechanism. The layer deposited impedes diffusion of the protons to the cathode that leads to increase of the protons concentration in the suspension and lower the pH.

2. It is also possible that the EPD process is associated with the desorption of potential-determining ions from the surface of nanoparticles during the formation of a precipitate, that leads to increase of the proton concentration in the suspension.

During the EPD, the zeta potential changed depending on the composition of the dispersion medium and the prehistory of the suspension. Freshly prepared suspensions both of pure isopropyl alcohol and with acetylacetone (YLO_i_1, YLO_iA_1) demonstrate increase in the zeta potential about 1.13–1.25 times after of the EPD.

Aging of YLO_iA_1 suspension for 2 months led to a significant decrease in the pH from 7.9 to 6.0 and zeta potential (by a factor of 1.66). During the EPD the YLO_iA_2 suspension demonstrated a higher average current (by a factor of 1.7). The zeta potential demonstrated little change, actually, +50 mV before the EPD and +49 mV after. It can be assumed that aging the suspension with acetylacetone leads to a slow change in the structure of the electrical double layer on the YLO nanoparticles.

The experiments performed showed that there is a relationship between the deposited nanopowders and the zeta potential (Figure 3). The Figure shows the general trend of increasing the weight of green bodies with the zeta potential increased. It should be noted that at the same zeta potential value of +24 mV (Figure 3), the weight of precipitates obtained by EPD from the YLO_i_1 suspension differs by a factor of 2 (188.6 mg for horizontal EPD and 91.4 mg for vertical EPD). It can be assumed that, during horizontal deposition, the EPD process is less stable than during vertical one due to the combination of the horizontal motion of the nanoparticles under the action of the field and the vertical component of hydrodynamic flows in suspension with constant stirring. Thus, we assume that horizontal deposition is associated with more intense mixing of the medium, which causes the formation of a large inhomogeneous green-body. It can be seen that in the range of zeta potential values from +50 to +80 mV, the mass deposited ceases to depend on the zeta potential value.

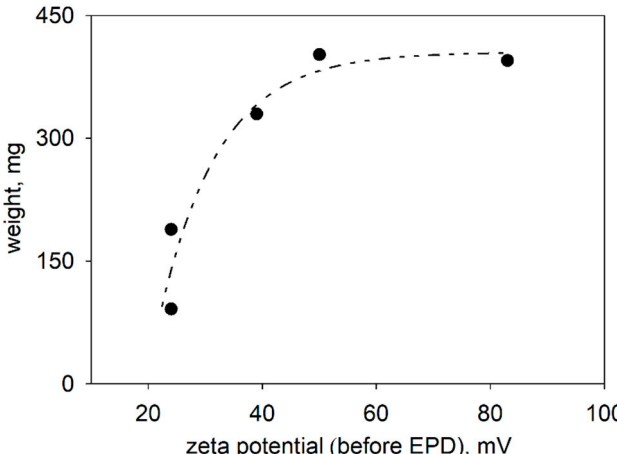

**Figure 3.** Dependence of weight of the deposited green body on the zeta potential of nanoparticles in the suspension.

The green-body growth kinetics curve (Figure 4) has an S-shaped character: at the initial period (up to 100 min) the mass growth rate increases, then the growth gradually slows down. The green body grows during the entire EPD process.

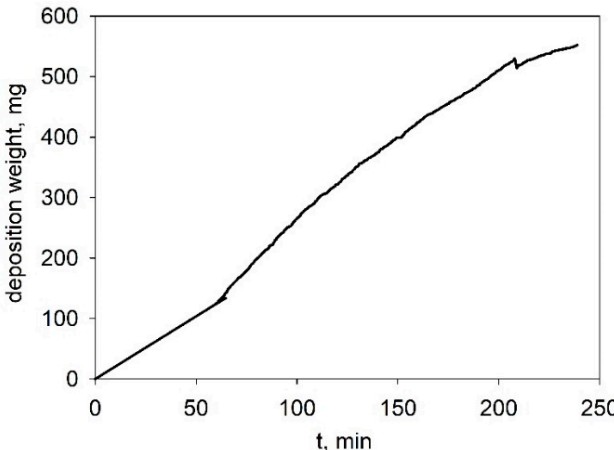

**Figure 4.** Kinetics of the green body growth vs. duration of the EPD from a YLO_iA_1 suspension at a constant electric field of 20 V/cm.

The energy consumption of this process, i.e., the ratio of the mass of the compact to the total charge that has passed through the suspension, are shown in Table 1. It can be seen that this ratio in the case of suspensions with acetylacetone (YLO_iA) is 4–7 times lower than without it (YLO_i). At the same time, the weight of the green bodies obtained from YLO_iA suspensions is 2–3 times higher. It is quite understandable, since the growth of the green body deposited on the electrode is caused solely by transfer of particles, and the charge transfer is due to both ions and particles movements, the charge passed through the suspension only partially characterizes the weight of the deposited material.

Figure 5 shows the dependences of the current on time at a constant electric field of 20 V/cm for vertical and horizontal EPD from YLO_i_1 suspension. In both cases, within a short time after the start of the EPD, there was a rapid drop in the current by 20–40%. Then, the process became different. With vertical deposition (Figure 5a), the current intensity during EPD first increases (~30 min), and then gradually decreases, reaching a plateau. During horizontal deposition (Figure 5b), there was a monotonous decrease in the current. The difference in the current kinetics can be due to the different nature of the hydrodynamic flows in the medium and their interactions with the flow of particles and with the layer deposited. The decrease in the current over time is possibly associated with an increase in

the resistance of the green body deposited and reduce of concentration of the nanoparticles ("depletion") in the suspension.

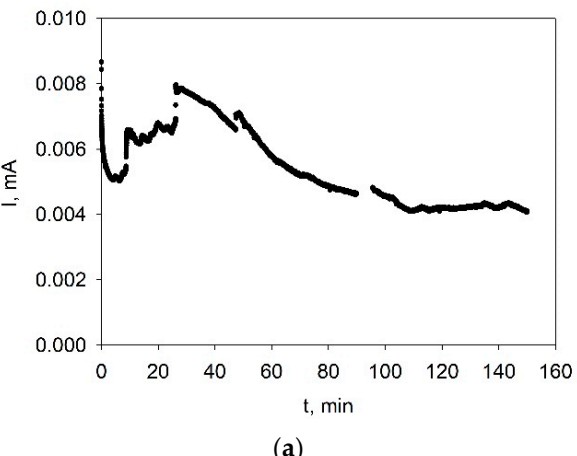

(a)

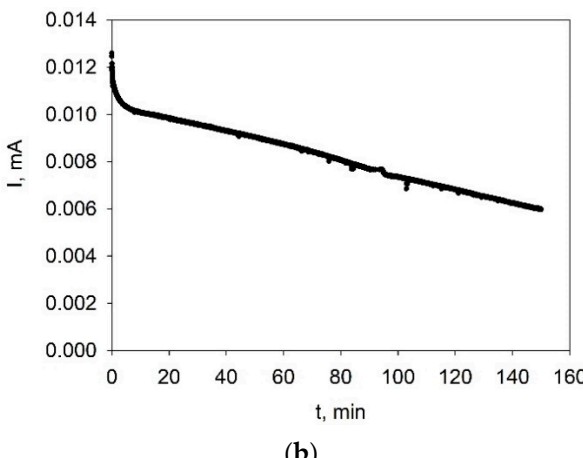

(b)

**Figure 5.** Time dependences of current at a constant electric field of 20 V/cm for (**a**) vertical and (**b**) horizontal EPD from YLO_i_1 suspension.

Figure 6 shows wet compacts immediately after EPD from YLO_i_1 suspension. In both cases, a coagulated gel-like green body are formed on the cathode. It can be seen that a uniform compact was formed in the cell during vertical deposition, in contrast to horizontal one, which is characterized by the formation of a compact with bumpy ridges along the edges of the cell. In the case of horizontal deposition (Figure 6b), the inhomogeneity of the compact surface can be associated with the redistribution of the particle concentration near the cathode under the action of gravity, which causes the formation of a green body that is inhomogeneous in thickness.

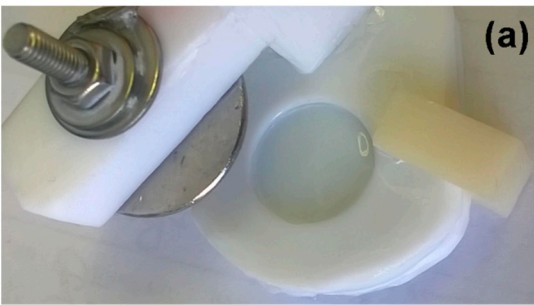
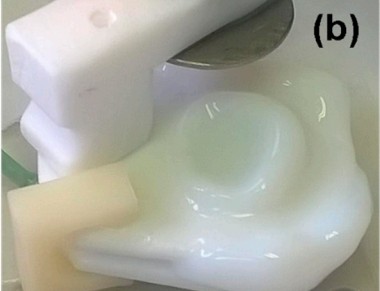

**Figure 6.** Picture of wet green bodies immediately after EPD in the case of (**a**) vertical and (**b**) horizontal deposition from YLO_i_1 suspension.

Figure 7 shows the dependence of the current on time during EPD at a constant electric field strength of 20 V/cm from a suspension with the addition of acetylacetone-YLO_iA_1. First of all, it should be noted that the current in the suspension with acetylacetone is an order of magnitude greater than in the suspensions with pure isopropanol. This fact is in good agreement with a significant (almost four times) difference in the zeta potential of nanoparticles in suspensions (Table 1). One can see an unambiguous tendency for the current to increase from 0.075 to 0.094 mA in the time interval from 0 to 150 min. The supposed reason is an increase in the conductivity of the liquid medium due to ion transfer with an increase in the concentration of $H^+$ ions as a result of the electrochemical reaction at the electrodes, which was described above. Worth mentioning is the fact that in the suspension there are actually two competitive processes: increase the resistance of the layer deposited and increase in the conductivity of the suspension due to an increase in the concentration of protons.

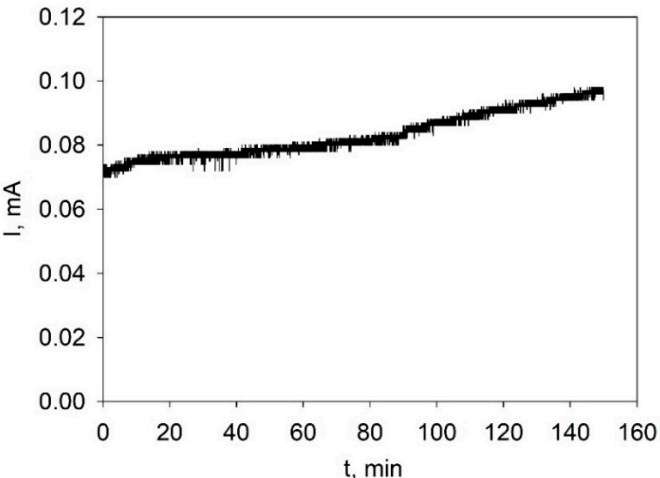

**Figure 7.** Dependence of the current on time during EPD at a constant electric field of 20 V/cm from a suspension with the addition of acetylacetone-YLO_iA_1.

A different picture was observed during EPD from a suspension aged for two months YLO_iA_2. There was a tendency to decrease the current (0.145–0.131 mA), approximately 1.11 times (Figure 8a). The difference in the kinetics during deposition from YLO_iA_1 (Figure 7) and YLO_iA_2 (Figure 8a) suspensions can be associated with a change in the effective particle charge, i.e., zeta potential and ionic composition of the suspension during the aging.

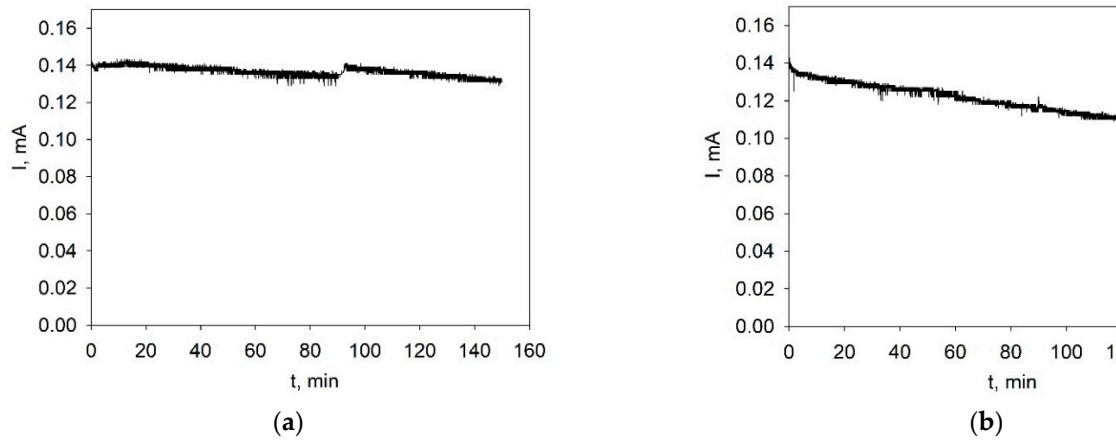

(**a**)                                                                                              (**b**)

**Figure 8.** Dependences of current on time during EPD at a constant electric field of 20 V/cm from suspensions: (**a**)-YLO_iA_2; (**b**)-YLO_iA_3.

In order to determine the changes that occur in the YLO_iA_2 suspension with a decrease in the concentration of nanoparticles during the EPD, a repeated EPD (YLO_iA_3 suspension) was carried out under the same modes (E= 20 V/cm, t = 150 min). Figure 8b shows the dependence of the current on time during the EPD from YLO_iA_3 suspension. It can be seen from Figure 8b that the current decreases linearly with time (0.147–0.106 mA). It should be noted that the difference in the current kinetics from the YLO_iA_2 (Figure 8a) and YLO_iA_3 (Figure 8b) suspensions is minimal, which indicates similar electrokinetic properties and the steady-state nature of the conductivity of these suspensions.

Table 2 presents the results describing the effect of the suspension used, the type and time of deposition on the thickness and density of the resulting green bodies. It's seen that addition of the acetylacetone leads to density of the compacts increased from 12 to 30% of theoretical. Longer deposition time leads to a large thickness of the compact but a lower density. Aged suspension as well as depleted one used for the EPD negatively affects the density of the resulting compacts. The highest density compact was obtained by vertical

deposition from the YLO_iA_1 suspension (Figure 9). Its mass was 395 mg, thickness was 2.4 mm, and density was 37%.

**Table 2.** Suspensions, EPD parameters, size and density of the green bodies.

| Suspension | EPD Direction and Time (min) | Thickness, mm | Density, % of Theoretical |
|---|---|---|---|
| YLO_i_1 | Vertical, 150 | 2 | 12 |
| YLO_i_1 | Horizontal, 150 | 4 | 17 |
| YLO_iA_1 | Vertical, 150 | 3.2 | 30 |
| YLO_iA_1 | Vertical, 90 | 2.4 | 37 |
| YLO_iA_2 aged | Vertical, 150 | 3 | 29 |
| YLO_iA_3 repeated EPD from YLO_iA_2 | Vertical, 150 | 3.5 | 25 |

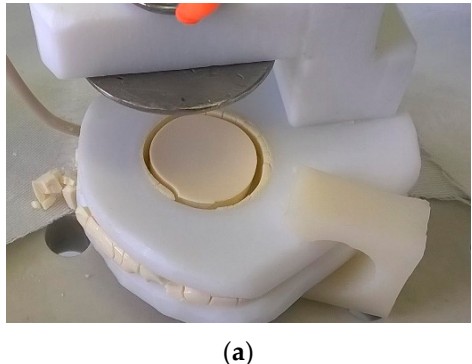 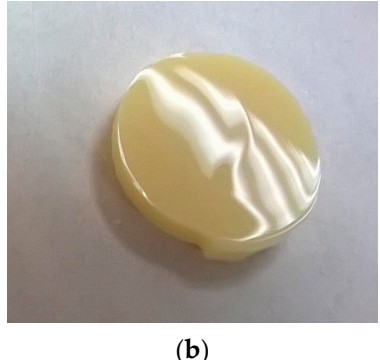

(**a**)      (**b**)

**Figure 9.** Dry compact obtained by vertical EPD from YLO_iA_1 suspension: (**a**)–at the beginning of the drying process in the cell; (**b**) dry compact.

We believe that in the case of the aged YLO_iA_2 and depleted YLO_iA_3 suspensions, the density of the compacts decreases due to a decrease in the zeta potential compared to the freshly prepared YLO_iA_1 suspension (from +83 mV to +50 and to +39 mV for YLO_iA_2 and YLO_iA_3 suspensions, respectively). Thus, the effective charge on the particles became lower, which reduced the force acting on them during the deposition from the external electric field, thereby reducing the green-body density.

## 3. Materials and Methods

### 3.1. Nanopowders and Suspensions

Nanoparticles of solid solution $(La_xY_{1-x})_2O_3$, x = 0.1 (YLO) were synthesized by laser ablation (LA) with an ytterbium fiber laser YLR-1000 (IPG Photonics, Fryazino, Russia,) $\lambda = 1.07$ μ. The material to evaporate was mixed from commercially available powders of high purity yttrium and lanthanum oxides (99.99% Polirit, Moscow, Russia). The laser ablation setup is described in detail in [31]. The laser operated at a pulse repetition rate of 5 kHz at a laser pulse duration of 60 ms and an average laser power of 255 W. Intensity of the laser radiation in focal spot was about 106 W/cm$^2$ with close-to-Gaussian profile. The oxides were ablated an air.

Specific surface area ($S_{BET}$) of the nanopowders was measured by nitrogen adsorption according to the BET method using a TriStar 3000 vacuum sorption unit (Micrometrics, Norcross, USA). Phase structure and composition of the nanopowders were characterized by X-ray diffraction (XRD) using D8 DISCOVER (Bruker AXS, Karlsruhe, Germany) with Cu (K$_\alpha$) radiation and a carbon monochromator. The XRD data were analyzed using TOPAS 3

software (Bruker AXS, Karlsruhe, Germany) with Ritveld's algorithm. Gas-phase products and exo/endothermal reactions during annealing of the samples at up to 1400 °C were analyzed by a thermogravimetric analysis (TGA) provided with the differential scanning calorimetry (DSC) with NETZSCH-STA409PC (NETZSCH, Selb, Germany) in argon at a heating rate of 10 °C/min. Morphology and particle sizes were observed by transmission electron microscopy (TEM) with JEM-2100 (JEOL, Tokyo, Japan). The particle size distribution by number was calculated from the pictures for at least 1500 particles. Particle size measurements using dynamic light scattering (DLS) were performed with a ZetaSizer Nano ZS system (Malvern Instruments, Malvern, United Kingdom). The parameters used were: refractive index $n(Y_2O_3) = 1.93$, $n$(2-Propanol) = 1.377, isopropanol viscosity 2.4 cP, T = 25 °C. To obtain suspensions of YLO nanopowder, isopropanol (special purity grade, JSC «Component-Reaktiv», Moscow, Russia) was used as a dispersion medium. Acetylacetone (analytically pure grade, Merck, Darmstadt, Germany) was used as a dispersant. Concentration of the acetylacetone was chosen based on the calculation of the mass of acetylacetone per total surface area of nanoparticles in suspension according to the formula:

$$m_{Hacac} = \mu \cdot S_{BET} \cdot m_{np}$$

where: $\mu$—mass of acetylacetone per unit surface of the nanopowder, mg/m$^2$,

$\quad$ $S_{BET}$—specific surface of the nanopowder, m$^2$/g;

$\quad$ $m_{np}$—weight of the nanoparticles, g;

$\quad$ $m_{Hacac}$—weight of the acetylacetone, mg

In accordance of a series of previous experiments on the stabilization of nanoparticles in suspensions [32,33], a value of $\mu$ about 1 mg/m$^2$ was chosen. Suspensions where acetylacetone was used are listed in Table 1.

Suspensions with an initial concentration of 70 g/L were prepared by accurately weighing the nanopowder and sonicated using an ultrasonic bath UZV-13/150-TN (Reltec, Yekaterinburg, Russia) for 125 min. Undestroyed large aggregates were removed by centrifugation with a Z383 centrifuge (Hermle Labortechnik GmbH, Wehingen, Germany) at a speed of 10,000 rpm for 3 min. The concentration of deaggregated suspensions was 62 g/L.

### 3.2. Suspension Characterization Methods and EPD

Electrokinetic measurements were performed by the electroacoustic method using a DT-300 analyzer (Dispersion Technology, Bedford Hills, NY, USA). pH measurement was carried out using a portable AS218 pH meter. It should be clarified that pH measurements were carried out in alcoholic suspensions; therefore, in the analysis the main attention was paid not to the absolute value of pH, but to its change during EPD, which serves as an indicator of changes in the concentration of protons in the medium. All measurements were carried out under isothermal conditions in air at 298 K.

Electrophoretic deposition was performed on a computerized setup providing constant voltage regimes, which was developed and manufactured at the Institute of Electrophysics of the Ural Branch of Russian Academy of Sciences. EPD was performed with two types of electrode arrangement: vertical and horizontal. With a vertical arrangement of electrodes, deposition is carried out in a horizontal direction, in the case of a horizontal arrangement of electrodes-vertical deposition. An aluminium foil disk with an area of 113 mm$^2$ served as a cathode, and a stainless-steel disk served as an anode; the distance between the electrodes was 10 mm. The EPD was made at a constant electric field of 20 V/cm. The deposition time was up to 180 min.

### 4. Conclusions

In the present work, a study was carried out to investigate the key factors that determine the uniformity, mass, thickness, and density of compacts obtained from nanopowders of solid solutions of yttrium and lanthanum oxides (($La_xY_{1-x})_2O_3$) with the help of the electrophoretic deposition (EPD). It has been shown that acetylacetone with a concentration

of 1 mg/m$^2$ can be used as a dispersant for stabilization of isopropanol suspensions of the nanoparticles during the EPD. It was found that depending on the presence of the dispersant and the prehistory of the suspension, the zeta potential of the nanoparticles changed during the EPD. With an increase in the zeta potential of the nanoparticles the weight of the material deposited increases. The deposition rate increases at the first stage of the EPD and then slows down with the course of the process. It was shown that the maximum density of dry compacts with a thickness of 2.4 mm reaches 37% of theoretical when EPD is performed in vertical direction from a suspension of nanopowders with addition of acetylacetone. The use of aged or depleted suspensions leads to a decrease in the compact density.

**Author Contributions:** Conceptualization, E.K. and M.I.; methodology, E.K. and M.I.; investigation, E.K.; resources, M.I.; data curation, E.K.; writing—original draft preparation, E.K.; writing—review and editing, M.I.; visualization, E.K. and M.I.; project administration, M.I.; funding acquisition, M.I. All authors have read and agreed to the published version of the manuscript.

**Funding:** This research was funded by the Russian Science Foundation, grant number 18-13-00355. The electron microscope of the Ural Center for Shared Use "Modern nanotechnology" Ural Federal University (Reg. No. 2968) was used with financial support of Ministry of Science and Higher Education of the RF (Project No. 075-15-2021-677). The laser ablation equipment of the Center for Shared Use "Electrophysics" in the Institute of Electrophysics, Ural Branch of Russian Academy of Sciences was used with financial support of Ministry of Science and Higher Education of the RF (No. 122011200363-9).

**Institutional Review Board Statement:** Not applicable.

**Informed Consent Statement:** Not applicable.

**Data Availability Statement:** Not applicable.

**Acknowledgments:** Maxim Ivanov would like to express deep gratitude to Michael Bredol (FH Münster, Germany) for his generous and helpful advices on EPD of nanoparticles.

**Conflicts of Interest:** The authors declare no conflict of interest.

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
