# Peer review of "The Electrophoretic Deposition of Nanopowders Based on Yttrium Oxide for Bulk Ceramics Fabrication"

_inorganics, doi:10.3390/inorganics10120243_

Round 1

Reviewer 1 Report

Dear Authors,

I find you paper interesting but there are some results that are missing.

In the experimental part, you talk about XRD along with Rietveld and TGA-DSC, but there are no such results.

Figure 1 and Figure 2. should be written in Latin instead of Cyrilic letter.

References should be all written in the same style, according to journal standards.

Everything else is good!

Reviewer 2 Report

The paper « Electrophoretic deposition of nanopowders based on yttrium oxide for bulk ceramics fabrication » assumes to give an original investigation on effect of process parameters on the preparation of (LaxY1-x)2O3 materials by EPD. Overall, the article has a clear purpose and is well organized. However, some phenomena need to be analyzed and explored in depth. Specific comments and suggestions are listed below:

1.     The authors describe the advantages, disadvantages and relevant principles of electrophoretic deposition methods in the INTRODUCTION section, however, there is less of an overview of current research progress and it is recommended to add descriptions of specific studies of related work to facilitate comparison by the reader.

2.     Lack of emphasis on innovative points in the article, despite being a process-related study (Line 72-80).

3.     If possible, please explain the faulting of the curves in Figures 5(a) and 8(a).

4.     The relationship between current and time before and after the addition of acetylacetone showed diametrically opposed results. Is there a competing mechanism between H+ concentration and green-body resistance? A comparison is suggested to explain the reasons for this shift (Line 296).

5.     It is recommended that the description of the phenomenon is supplemented by the corresponding analysis and explanation. For example, the reason why “Aged suspension as well as depleted one used for the EPD negatively affects the density of the resulting compacts” (Line 326).

Reviewer 3 Report

This paper is a clear and interesting study on electrophoretic deposition (EPD) of nanopowders for ceramic applications. Please consider the remarks below before publication:

-      -    Introduction should enhance better the difficulty of nanopowders forming, even through a colloidal process. The low density obtained for all the samples is linked with the high specific surface area of the powders, and it should be a key aspect of the study.

-   In the experimental part, please precise:

o   Why such a high heating rate for TGA (40°C/min)?

o   How is obtained the size distribution in Figure 1b?

o   In which solvent and at which concentration is measured zeta potential of the suspension? I’m a bit surprised by the very high values measured if isopropanol is used, please comment

o   Dynamic light scattering is mentioned L.176, but not in the experimental methods, please add the equipment and conditions of measurements

-  L.163 – L.166: I don’t agree with this reaction mechanism. If the adsorption of acetylacetone at the particle surface increases zeta potential, it’s through a protonation mechanism of the surface, and not through the formation of free protons. Maybe there is a complexation with this molecule, I would search for another mechanism in this direction

-    EPD is very impacted by the viscosity of the suspension. Do you have any measurement of the viscosity of the suspension before deposition? The deposit is a gel-like material, is the rest of the suspension similar, or is it remain fluid?

- All the compacts present low density after drying, the highest value being 37%. Did the authors study the thermal behavior of such compacts, and their sinterability? Did they obtain bulk ceramics? It is important to comment on this aspect, to be coherent with the title of the paper.

-  Small errors to correct:

o   L.65 what is EPT, or do you mean EPD ?

o   L.127, L.212 : g/l should be changed in g/L

-  Excessive auto-citations: 16 citation on 30 are from one of the 3 authors

Round 2

Reviewer 1 Report

Dear authors,

you made the corrections and improved your paper. I suggest publishing!

Best regards